# Delayed Onset of Thrombotic Microangiopathy (TMA) upon Prolonged Carfilzomib Therapy in Multiple Myeloma: A Case Report and Comprehensive Review

**DOI:** 10.3390/ph17121722

**Published:** 2024-12-20

**Authors:** Andrea Ceglédi, Ágnes Király, Andrea Várkonyi, Szabolcs Tasnády, Hajnalka Andrikovics, Mónika Fekete, Bálint G. Szabó, Zsuzsanna Szemlaky, Ágnes Szilágyi, György Sinkovits, Zoltán Prohászka, Marienn Réti, Gábor Mikala

**Affiliations:** 1Department of Hematology and Stem Cell Transplantation, South Pest Central Hospital, National Institute of Hematology and Infectious Diseases, 1097 Budapest, Hungary; ceglediandi@freemail.hu (A.C.); kiraly.agnes@dpckorhaz.hu (Á.K.); andreavarkonyi80@gmail.com (A.V.); sz.tasnady@gmail.com (S.T.); szabo.balint.gergely@dpckorhaz.hu (B.G.S.); szemlaky.zsuzsanna@dpckorhaz.hu (Z.S.); retimarienn@gmail.com (M.R.); 2Health Sciences Program, Doctoral College, Semmelweis University, 1085 Budapest, Hungary; 3Institute of Preventive Medicine and Public Health, Semmelweis University, 1085 Budapest, Hungary; andrikovics.hajnalka@dpckorhaz.hu (H.A.); feketemonika78@gmail.com (M.F.); 4Laboratory of Molecular Genetics, Central Hospital of Southern Pest, National Institute of Hematology and Infectious Diseases, 1097 Budapest, Hungary; 5Department of Internal Medicine and Hematology, Füst György Complement Diagnostic Laboratory, Semmelweis University, 1088 Budapest, Hungary; szilagyi.agnes@semmelweis.hu (Á.S.); sinkovits.gyorgy@semmelweis.hu (G.S.); prohaszka.zoltan@semmelweis.hu (Z.P.)

**Keywords:** thrombotic microangiopathy, carfilzomib, multiple myeloma, plasmapheresis, delayed onset

## Abstract

Background: Thrombotic microangiopathy (TMA) is a potentially life-threatening complication associated with carfilzomib, a proteasome inhibitor approved for treating multiple myeloma. TMA typically presents within the initial months of treatment; however, delayed onset is rare and poses significant diagnostic challenges. Methods: We conducted a retrospective analysis of the medical records of a 47-year-old Caucasian woman diagnosed with IgA kappa myeloma who developed signs and symptoms consistent with TMA eleven months after the initiation of carfilzomib therapy and already in ongoing very good partial remission. Results: The clinical presentation included an acute onset of weakness, dizziness, somnolence, diffuse bruising, oliguria, jaundice, severe thrombocytopenia, and acute kidney injury. An immediate workup raised a strong suspicion for TMA, confirmed by laboratory findings of schistocytosis and complement activation. Following the immediate discontinuation of carfilzomib, the patient underwent 18 plasmapheresis (PEX) sessions and received supportive fresh frozen plasma transfusions, which resulted in the complete remission of TMA symptoms without the need for complement inhibitory therapy. Conclusions: The need for ongoing monitoring for TMA throughout carfilzomib therapy, regardless of treatment duration, is emphasized. Early diagnosis and intervention, including drug discontinuation and the timely initiation of PEX, are crucial for patient recovery.

## 1. Introduction

Thrombotic microangiopathy (TMA) encompasses a spectrum of disorders characterized by microangiopathic hemolytic anemia, thrombocytopenia, and organ dysfunction due to microvascular thrombosis [1,2,3]. Among the therapeutic agents implicated in TMA [4,5,6,7,8], carfilzomib, a proteasome inhibitor approved for the treatment of multiple myeloma, has been identified as a potential causative agent [9,10,11]. Carfilzomib-induced TMA [12,13,14,15,16,17,18,19] is a serious but rare adverse effect that typically presents within the first months of treatment initiation [20,21]. However, cases with a delayed onset, occurring several months after therapy commencement, are particularly rare and can complicate clinical diagnosis and management due to their atypical presentation [7,22].

Multiple myeloma is a plasma cell neoplasm that often necessitates treatment with proteasome inhibitors, among which carfilzomib, an irreversible inhibitor, has been shown to offer significant therapeutic benefits [23,24,25,26]. The pathogenesis of the rare carfilzomib-induced TMA remains not fully understood, though it is hypothesized to involve endothelial injury and the subsequent activation of the coagulation cascade [15].

In this paper, we present a case of a 47-year-old Caucasian woman with IgA kappa myeloma who developed TMA following 11 months of carfilzomib treatment, while in near complete remission. This case is of particular interest due to the delayed onset of TMA, posing a diagnostic challenge and necessitating a prompt and effective treatment approach. Through this report, we aim to highlight the importance of considering TMA in carfilzomib-treated patients—especially if they are thrombocytopenic—regardless of the treatment duration, and the need for vigilant monitoring to ensure early detection and intervention.

## 2. Case Presentation and Results

A 47-year-old woman, with a history of lower extremity varicose veins, presented in the spring of 2019 with two months of persistent back pain, gait disturbance, and severe diffuse lower limb sensory deficits. An MRI of the thoracic spine revealed a tumor originating from the ThIII vertebra, extending dorsally and compressing the spinal cord. Surgical intervention on 20 April 2019, involved ThIII microdecompression and stabilization from ThII to ThIV. A histological examination of the surgical sample confirmed atypical plasma cell infiltration, with immunohistochemistry showing strong CD138, weak CD38, and focal CD56 positivity, and a Ki67 index of approximately 4%, leading to a diagnosis of plasmacytoma.

In May 2019, further evaluations confirmed IgA kappa multiple myeloma (ISS stage I) with high-risk cytogenetic features, including translocation t (4;14), monosomy of chromosome 13, and TP53 deletion. First-line treatment with the VDT protocol (bortezomib, thalidomide, and dexamethasone) achieved complete remission after three cycles. Following the fourth cycle, high-dose cyclophosphamide and G-CSF were used for peripheral stem cell mobilization and collection. The patient then received a fifth cycle of VDT therapy before undergoing autologous stem cell transplantation (ASCT) with melphalan conditioning on 22 November 2019.

Post-transplant maintenance therapy with lenalidomide began on 24 April 2020, but was discontinued on 3 July 2020, due to an extensive maculopapular skin rash. In January 2022, a biochemical relapse of myeloma was detected, and progressive disease prompted the initiation of PdV (pomalidomide, dexamethasone, and bortezomib) therapy on 13 May 2022. This regimen was also discontinued due to a pomalidomide-induced skin rash. On 19 July 2022, the treatment was switched to KD-bendamustine (carfilzomib, dexamethasone, and bendamustine) (Figure 1).

During the first cycle of carfilzomib, the patient developed steroid-induced diabetes, but no other significant complications were observed while her myeloma reached very good partial remission. Carfilzomib was administered bi-weekly from 17 January 2023 (bendamustine was also terminated), with only minor neutropenia managed with G-CSF until 23 May 2023. On 27 May 2023, the patient experienced acute symptoms of weakness, dizziness, somnolence, diffuse suffusions, decreased urine output, jaundice, severe thrombocytopenia, and impaired renal function, leading to urgent hospitalization.

Clinical findings and the peripheral smear results (presence of schistocytes) strongly suggested TMA. The patient’s laboratory values at the time of presentation are detailed in Table 1 and Table 2.

Laboratory tests indicated that ADAMTS13 activity was only moderately reduced (58% on 26 May and 45% on 30 May; reference range 67–151%), not to the deficient levels typical of thrombotic thrombocytopenic purpura (TTP), effectively ruling out TTP. Complement evaluation showed a decrease in complement activity (total classical pathway hemolytic activity 86 CH50/mL on 26 May and 44 CH50/mL on 30 May; reference range 48–103 CH50/mL), a decrease in complement C3 (1.11 g/L on 26 May to 0.66 g/L on May 30; reference range 0.9–1.8 g/L), and a marked decrease in complement C4 (0.32 g/L on May 26 to 0.03 g/L on 30 May; reference range 0.15–0.55 g/L), indicative of complement consumption. The presence of elevated terminal complement complex (SC5b-9: 554 ng/mL on 26 May; 411 ng/mL on 30 May, reference range 110–252 ng/mL) suggested the presence of complement activation behind consumption. Factor I and H levels and alternative pathway activity remained within the normal range, which did not support the presence of complement dysregulation. Despite mild upper respiratory symptoms, no concurrent infection could be confirmed (Table 3).

We measured von Willebrand factor (VWF) and ADAMTS13 activity from blood samples drawn just before the first plasma exchange (PEX). The results showed a VWF antigen level of 408%, a VWF ristocetin cofactor activity (VWF/RCo) of 219%, and a VWF collagen-binding activity (VWF/CBA) of 462%, with an ADAMTS13 activity at 58%. These values resulted in a markedly elevated VWF–Ag/ADAMTS13–Ac ratio of seven and a VWF–CBA/ADAMTS13–Ac ratio of eight, which were significantly higher than typical levels. The observed VWF–Ag level and VWF–Ag/ADAMTS13–Ac ratio were similar to those reported in a cohort of 53 acute TMA patients with coexisting conditions such as malignancy, sepsis, heart surgery, solid-organ transplantation, and systemic autoimmune disorders, as described in our previous study [27]. These findings further supported the diagnosis of TMA and underscored the complex interplay of endothelial activation, von Willebrand factor elevation, and moderate ADAMTS13 deficiency in this case. The complement consumption observed in parallel reinforced the hypothesis of a multifactorial etiology involving complement activation and endothelial injury, rather than isolated TTP or complement dysregulation alone.

A genetic analysis of the patient did not reveal pathogenic variants in any of the atypical hemolytic-uremic-syndrome-(aHUS)-related genes tested (*CFH*, *CFI*, *CD46*, *C3*, *CFB*, and *THBD*). Additionally, the multiplex ligation-dependent probe amplification of the CD46 gene did not show any deletions or duplications. The MCPggaac haplotype, associated with lower transcriptional activity, was also not identified, which could have explained the low MCP levels detected in the patient.

Based on these findings, carfilzomib was immediately discontinued, and plasmapheresis (PEX) was initiated. High-volume PEX was performed daily upon suspicion of thrombotic microangiopathy (TMA). The exchange volume was calculated as a median of 1.4 times the plasma volume per session (range: 1.3–1.5 times plasma volume), with substitution fluid comprising 500 mL of 4% human albumin combined with 14 units of fresh frozen plasma (FFP) per session. (In Hungary, FFP is the standard substitution fluid for TTP/aHUS/TMA indications. Although TRALI-safe Octaplas is theoretically available, its very high cost is not reimbursed, precluding its routine use.) PEX was performed daily for a total of 18 consecutive sessions. In the initial phase, no additional PEX sessions were conducted on the same day, but extensive plasma exchange was employed from the outset. A complete TMA response was observed after 15 daily PEX sessions, as evidenced by the normalization of the platelet count, lactate dehydrogenase (LDH), and haptoglobin levels, along with a >25% decrease in serum creatinine levels. PEX therapy was extended by an additional three daily sessions while awaiting insurance approval for complement inhibitory therapy, bringing the total to 18 sessions. (Figure 2) PEX was discontinued without significant adverse events after the patient achieved clinical remission. A request for complement inhibitory therapy (eculizumab) insurance approval was submitted early in the treatment process. However, by the time insurance approval was granted—one week after PEX was stopped—the patient had already achieved clinical and laboratory remission (normal platelet count, LDH, haptoglobin, and sC5b-9 levels; estimated glomerular filtration rate [eGFR]: 55 mL/min). At that point, complement inhibitory therapy was deemed unnecessary, as there were no signs of ongoing TMA activity. The efficacy of PEX was monitored using a combination of platelet counts, LDH levels, haptoglobin levels, and renal function markers (e.g., serum creatinine). Complement activity (sC5b-9) levels were also measured. An unsatisfactory PEX response would have been defined as failure to achieve the normalization of or improvement in these key laboratory and clinical markers within a reasonable timeframe. In this case, the observed rapid and sustained improvement indicated a satisfactory response. Importantly, throughout this period, the patient’s myeloma remained in remission, as evidenced by stable immunoglobulin levels and the absence of clinical relapse.

This case highlights the importance of prompt diagnosis and intervention in managing carfilzomib-induced TMA, emphasizing the need for ongoing monitoring and vigilance throughout treatment. Our case is noteworthy because TMA developed nearly one year into carfilzomib therapy, a rare occurrence that underscores the necessity for continuous vigilance throughout the entire treatment duration. The prognosis of drug-induced TMA hinges critically on the immediate suspension of the provoking drug, in this case, carfilzomib, and the prompt initiation of PEX treatments. Therefore, it is essential to consider TMA whenever thrombocytopenia arises during carfilzomib therapy, not just at its onset but throughout its duration. This is an intricate process, as proteasome inhibitors are well-known to cause transient reversible thrombocytopenia through the inhibition of megakaryocyte budding. While recent therapeutic successes with the terminal complement inhibitor eculizumab have been reported, in our patient, the rapid diagnosis followed by the immediate initiation of plasmapheresis alone resulted in complete remission. This case underscores the potential effectiveness of timely plasmapheresis in achieving full remission without the need for additional therapies such as eculizumab.

## 3. Discussion

Multiple myeloma represents a multifaceted hematologic malignancy marked by the clonal expansion of plasma cells within the bone marrow. This disease predominantly affects the elderly, with a median age at diagnosis of around 70 years, underscoring its classification as an age-associated disease [28]. The prevalence of multiple myeloma escalates with age, positioning it as a more common condition among the older population [28]. In recent years, the landscape of multiple myeloma treatment has witnessed substantial advancements, markedly enhancing patient outcomes. Among these advancements, carfilzomib has emerged as a pivotal treatment option, offering significant efficacy in the management of multiple myeloma [23,24,25,26,29,30,31,32,33,34,35,36,37].

Carfilzomib, a second-generation proteasome inhibitor, operates by selectively and irreversibly inhibiting the activity of the 20S proteasome in plasma cells. The proteasome is a crucial enzyme complex responsible for degrading misfolded or damaged proteins, which are common in rapidly dividing cells, such as the malignant plasma cells that tend to retain protein secretory potential in multiple myeloma. By blocking the proteasome’s function, carfilzomib leads to an accumulation of toxic misfolded proteins within the cells, thereby inducing apoptosis, or programmed cell death, in the cancerous plasma cells. This mechanism of action is particularly effective against multiple myeloma, where the proliferation of abnormal immunoglobulin-secreting plasma cells is a hallmark of the disease. Clinical trials have demonstrated the effectiveness of carfilzomib, showing significant improvements in progression-free survival and overall response rates when used either as a single agent or in combination with other therapies such as corticosteroids and immunomodulatory drugs [23,24,25,26]. Its role in the treatment algorithm of multiple myeloma, particularly for patients who have relapsed or are refractory to first-line treatments, highlights the importance of carfilzomib in the current therapeutic arsenal against myeloma.

TMA associated with carfilzomib treatment is a rare but severe complication with critical implications for patient management and clinical outcomes [15,16,17,18,19]. The epidemiology of carfilzomib-induced TMA is difficult to characterize due to underreporting and the variability of clinical presentations. In clinical trials investigating carfilzomib for multiple myeloma, the incidence of drug-associated TMA varied [17,24]. While a phase II study of carfilzomib as a single agent did not report any cases of TMA, it did note significant rates of anemia (46%), thrombocytopenia (39%), and acute kidney injury (AKI) (1.5%) [24]. The ENDEAVOR trial, a phase III comparison of carfilzomib and dexamethasone versus bortezomib and dexamethasone, identified TMA in two of 463 patients [36]. Another phase II trial, the CARDAMON study, observed TMA in eight of 281 patients [17]. Fotiou et al. also documented TMA in 5% (6 out of 114) of patients treated with carfilzomib for relapsed and refractory multiple myeloma, highlighting the drug’s potential to induce TMA [38]. Collectively, the available data suggest that TMA occurs in a minority of patients, with reported cases highlighting both early and delayed onset following therapy initiation [4,5,6,7,8,9,10,11,12,13,14,15,16,17,18,19]. It is crucial to distinguish carfilzomib-induced TMA from other forms of TMA such as atypical hemolytic uremic syndrome (aHUS) and thrombotic thrombocytopenic purpura (TTP), which have different treatment implications [15].

In this context, our findings complement and extend the current understanding of carfilzomib-induced TMA, such as the comprehensive review by Fang et al. [21], which analyzed early-onset carfilzomib-induced TMA cases. While Fang et al. primarily emphasized early manifestations within the initial months of therapy [21], our study highlights the diagnostic and therapeutic challenges associated with delayed-onset TMA, occurring nearly one year into treatment. This temporal difference underscores the importance of continuous vigilance throughout carfilzomib therapy. Furthermore, our report provides a detailed examination of complement activation and endothelial injury, as evidenced by elevated SC5b-9 levels and a markedly high VWF–Ag/ADAMTS13–Ac ratio, contributing additional mechanistic insights beyond what has been previously reported.

Although the exact pathomechanism of carfilzomib-induced TMA is not entirely elucidated, it is postulated to involve endothelial damage [18,39,40,41], which triggers a cascade of events leading to microvascular thrombosis, hemolytic anemia, and organ injury. Proteasome inhibitors like carfilzomib may directly induce endothelial impairment [40,41], which, in conjunction with factors such as high-dose regimens, predisposing patient factors (e.g., genetic susceptibilities, pre-existing cardiovascular disease, etc.), or concomitant medications, might tip the balance towards a pro-thrombotic state [14,15,42]. The role of complement overactivation in carfilzomib-induced TMA is an area of ongoing investigation [14,15,18,42]. The ubiquitin–proteasome pathway, critical for protein degradation and homeostasis, when inhibited, is believed to cause dysregulation of the complement system. Such dysregulation manifests as excessive complement activation, a process that directly contributes to endothelial damage [18]. Complement-mediated endothelial damage is a well-recognized pathophysiological mechanism in aHUS, and emerging evidence suggests a similar process may occur in drug-induced TMA [14,15,18,42]. Variants in genes regulating the complement system may predispose certain individuals to TMA, suggesting a genetic susceptibility component [42]. The resulting endothelial injury is a key precursor to the formation of microvascular thrombi, a hallmark of TMA, which further leads to consumptive thrombocytopenia, hemolytic anemia, and consequent end-organ damage [18]. The presence of complement activation in our patient lends support to this hypothesis, although further research is needed to fully understand the underlying mechanisms and their therapeutic implications. Potential triggers for TMA include infection, systemic inflammation, or immune-mediated reactions, all of which may contribute to endothelial injury. In the case of carfilzomib, additional triggers may include dose-related toxicity or individual patient vulnerability to the drug’s effects on endothelial cells [14,15,42].

Managing TMA in the context of carfilzomib therapy requires a high index of suspicion and a proactive approach. The immediate discontinuation of carfilzomib is imperative upon the diagnosis of TMA, as continued exposure to the drug may exacerbate the condition. Plasmapheresis, as demonstrated in our case, can be effective in removing circulating factors contributing to TMA, such as autoantibodies, cytokines, or other mediators of endothelial damage [5,43].

In managing TMA associated with carfilzomib therapy, where complement activation plays a crucial role in pathogenesis, eculizumab presents a targeted therapeutic approach. Eculizumab, a humanized monoclonal antibody, acts by inhibiting the cleavage of complement protein C5, thereby preventing the downstream formation of the membrane attack complex (MAC) and the generation of the proinflammatory peptide C5a. This mechanism directly addresses the excessive complement activation that underlies the endothelial damage leading to TMA. Given the established role of complement dysregulation in the development of carfilzomib-induced TMA, eculizumab offers a rational therapeutic option [7,13,19], especially in cases where plasmapheresis alone does not suffice or when the rapid progression of TMA necessitates immediate intervention to prevent irreversible organ damage [44,45,46]. Our patient’s rapid diagnosis and immediate plasmapheresis resulted in complete remission without the need for eculizumab.

## 4. Materials and Methods

A retrospective chart review was performed for a 47-year-old Caucasian woman diagnosed with IgA kappa myeloma, who developed thrombotic microangiopathy (TMA) following treatment with carfilzomib. Ethical approval for data collection was granted by the Ethics Committee of the South Pest Central Hospital/National Institute of Hematology and Infectious Diseases, and the patient provided written informed consent for her data to be used in this publication.

The patient’s medical records provided a comprehensive source of data, including detailed laboratory test results, radiologic findings, pathology reports, and notes on the administration of medications. The timeline of the patient’s clinical progress was meticulously reconstructed from the initiation of carfilzomib treatment to the diagnosis and management of TMA, and up to the latest follow-up.

The evolution of the patient’s condition was charted, with a specific focus on the onset of TMA symptoms, laboratory abnormalities, and the response to therapeutic interventions, including the cessation of carfilzomib and the implementation of plasmapheresis (PEX). The outcome measures were the resolution of TMA symptoms and normalization of laboratory parameters.

The limitations of this analysis were inherent in its retrospective nature, including the potential for incomplete data and the challenges associated with inferring causation from clinical observations. Additionally, as a single case study, the findings may not be widely generalizable to all patients undergoing carfilzomib therapy but do contribute to the growing body of evidence regarding the management of carfilzomib-induced TMA.

## 5. Conclusions

Carfilzomib-induced TMA is a rare but serious complication that requires prompt recognition and intervention. Our case underscores the importance of considering TMA as a potential complication not only at the outset of therapy but throughout the entire duration of carfilzomib treatment. The identification of specific risk factors, understanding of pathomechanisms, and development of targeted treatment strategies will be instrumental in improving patient outcomes. As the use of carfilzomib continues to expand in multiple myeloma, further research is needed to establish evidence-based guidelines for the prevention, early detection, and management of TMA. Collaborative efforts to report new cases and share clinical experiences will enrich the body of literature and support healthcare professionals in delivering optimal care to patients at risk for this complication.

## Figures and Tables

**Figure 1 pharmaceuticals-17-01722-f001:**
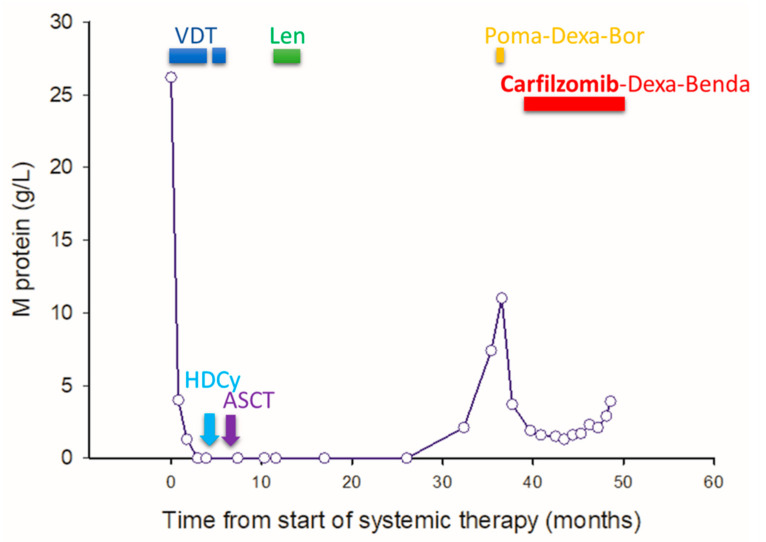
Time course of therapy-induced changes in M protein in a 47-year-old myeloma patient. The graph illustrates sequential serum concentrations of the M protein throughout the patient’s disease course, emphasizing the impact of various treatment regimens on disease progression and response. The treatments are annotated at the corresponding time points to delineate their duration and sequence. VDT refers to a regimen combining bortezomib, dexamethasone, and thalidomide, designed to target multiple pathways in myeloma cells. Len indicates treatment with lenalidomide, an immunomodulatory drug, which was discontinued due to cutaneous adverse effects. HDCy stands for high-dose cyclophosphamide, a chemotherapy regimen used to mobilize stem cells before autologous stem cell transplantation (ASCT). Poma-Dexa-Bor represents a combination therapy of pomalidomide, dexamethasone, and bortezomib, aimed at patients with relapsed multiple myeloma, which was discontinued due to adverse cutaneous side effects. Instead, carfilzomib, dexamethasone, and bendamustine (Carfilzomib-Dexa-Benda), which target proteasomes and DNA replication simultaneously in myeloma cells, were used in our patient. This figure encapsulates the therapeutic journey of the patient, highlighting the dynamic nature of multiple myeloma management and the tailored approach to treatment.

**Figure 2 pharmaceuticals-17-01722-f002:**
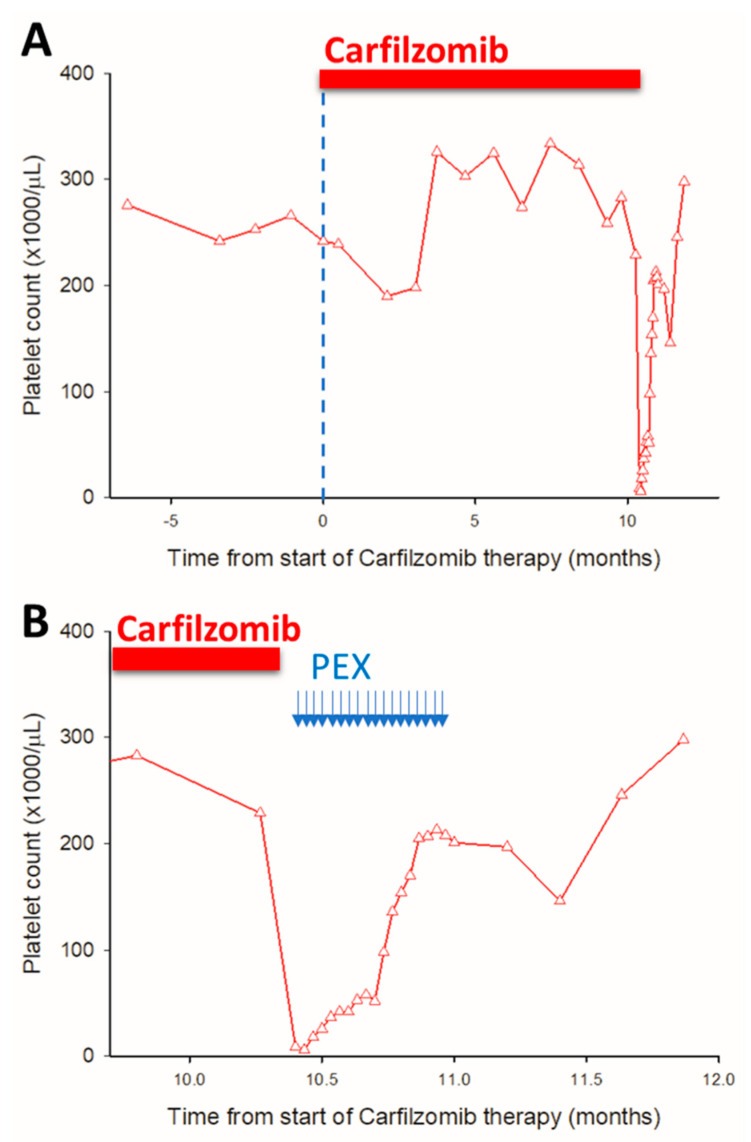
Time course of changes in platelet count in response to carfilzomib treatment in a 47-year-old myeloma patient. This figure is divided into two panels to detail the evolution of platelet counts over the course of carfilzomib therapy and subsequent interventions. (**Panel A**) displays the complete time course, illustrating that there were no significant changes in platelet counts during the initial 11 months of carfilzomib treatment, indicating a stable phase without apparent hematologic toxicity attributable to the therapy. (**Panel B**) focuses on the critical period when a sudden drop in platelet counts occurred, signaling the onset of TMA. Carfilzomib treatment was immediately discontinued in response to the TMA diagnosis. Following the cessation of carfilzomib, 18 sessions of plasmapheresis (PEX) were administered, which are depicted on the timeline. The successful completion of PEX sessions was closely correlated with the complete remission of TMA symptoms and the normalization of platelet counts, highlighting the effective management and resolution of this serious complication. This graphical representation underscores the importance of vigilant monitoring for hematologic adverse events. (PEX: plasma exchange therapy, vertical arrows represent individual PEX sessions).

**Table 1 pharmaceuticals-17-01722-t001:** Laboratory values at the outset of clinical symptoms.

Parameter	Value	Unit	Reference Range
Serum Sodium	140.0	mmol/L	135.0–145.0
Serum Potassium	3.80	mmol/L	3.50–5.60
Serum Calcium	2.30	mmol/L	2.08–2.65
Serum Magnesium	1.00	mmol/L	0.53–1.11
Serum Iron	27.5	µmol/L	6.3-26.9
Serum Phosphate	1.32	mmol/L	0.78–1.65
Blood Glucose	4.90	mmol/L	3.50–5.50
Total Bilirubin	131.3	µmol/L	5.0–21.0
Direct Bilirubin	11.3	µmol/L	<5.1
GOT (AST)	102	U/L	13–40
GPT (ALT)	53	U/L	7–40
Gamma GT	123	U/L	8–42
Alkaline Phosphatase	71	U/L	35–105
LDH	4343	U/L	240–480
Pseudo-Cholinesterase	9492.0	U/L	3700.0–13,200.0
Serum Amylase	33	U/L	<99
Lipase	43	U/L	12–53
CK	469	U/L	24–170
CKMB	21	U/L	<24
hs Troponin I	0.1570	ng/mL	<0.0155
Serum Urea	19.4	mmol/L	3.0–8.0
Serum Creatinine	200	µmol/L	44–97
eGFR (CKD-EPI)	25	mL/min/1.73 m²	>90
Total Serum Protein	58	g/L	57–82
Albumin	37	g/L	32–48
Cholesterol	5.6	mmol/L	<5.2
Triglyceride	3.0	mmol/L	0.9–1.7
HDL Cholesterol	1.30	mmol/L	>1.10
LDL Cholesterol	3.54	mmol/L	<3.40
Uric Acid	451	µmol/L	181–353
TSH	3.709	µIU/mL	0.350–4.940
Homocysteine	15.5	µmol/L	4.3–11.1
ADAMTS13 Metalloprotease Activity	58	%	67–151
Total Complement (Hemolytic Test)	86	CH50/mL	48–103
Alternative Pathway Total Complement (WIELISA-ALT)	114	%	70–125
Complement C3	1.11	g/L	0.9–1.8
Complement C4	0.32	g/L	0.15–0.55 g/L
H Factor Antigen Measurement	589	mg/L	250–880 mg/L
I Factor Antigen Measurement	101	%	70–130%
B Factor Antigen Measurement	124	%	70–130%
C1q Antigen	73	mg/L	60–180 mg/L
Anti-H Factor IgG Autoantibody	10	AU/mL	<110 AU/mL
Anti-C1q IgG Autoantibody	1	U/mL	<52 U/mL
SC5b-9 (Terminal Complement Complex)	554	ng/mL	110–252 ng/mL
Haptoglobin	<0.07	g/L	0.3–2.0 g/L
Alpha 1 Globulin	3.3	g/L	2.1–3.5
Alpha 2 Globulin	6.3	g/L	5.1–8.5
Beta 1 Globulin	2.7	g/L	3.4–5.2
Beta 2 Globulin	2.6	g/L	2.3–4.7
Gamma Globulin	7.3	g/L	8.0–13.5
IgG	2.96	g/L	7.00–16.00
IgA	5.89	g/L	0.90–4.50
IgM	<0.20	g/L	0.60–2.80
Free κ Light Chain	19.7	mg/L	6.7–22.4
Free λ Light Chain	8.88	mg/L	8.3–27
κ/λ Ratio	2.21		0.26–1.65
M Protein	2.9	g/L	
CRP	9.4	mg/L	<10.0
Rheumatoid Factor	6.00	U/mL	<14.00
CA 125	11.1	U/mL	<35.0
CA 15–3	9	U/mL	<28
CA 19–9	4	U/ mL	<39
ANA HEp-2 1:160 Titer	Negative		
Anti-Cytoplasm Antigen Antibody Detection	Negative		
Anti-Centromere Antibody	Negative		
Anti-PCNA Antibody	Negative		
Anti-dsDNA Antibody Determination	Negative		
ENA Screening	Negative		
ANCA IIF	Negative		
p-ANCA	Negative		
c-ANCA	Negative		
Atypical ANCA	Negative		

**Table 2 pharmaceuticals-17-01722-t002:** Coagulation tests at the time of TMA presentation.

Parameter	Value	Unit	Reference Range
Prothrombin Time	10.5	sec	9.4–12.5
INR	0.91	o	0.90–1.15
Activated Partial Thromboplastin Time (APTT)	27.9	sec	28.0–40.0
Thrombin Time	15.9	sec	10.3–16.6
Fibrinogen Derivative	5.11	g/L	2.76–4.71
D-Dimer	1901	ng/mL	<500
Factor VIII	377	%	50–150
Antithrombin Activity	97	%	83–128
Protein C	92	%	70–140
Protein S	99	%	64–149
Lupus Anticoagulant	Negative		

**Table 3 pharmaceuticals-17-01722-t003:** Virology and serology test results.

Test Description	Result
Treponema pallidum total antibody detection CLIA (DiaSorin Liaison XL)	Negative
Anti-Toxoplasma gondii IgM antibody detection CLIA (DiaSorin Liaison XL)	Negative
Anti-Toxoplasma gondii IgG antibody detection CLIA (DiaSorin Liaison XL)	Positive
HIV-1 antigen and antibody, HIV-2 antibody detection CLIA (DiaSorin Liaison XL)	Negative
Anti-HCV antibody detection CLIA (DiaSorin Liaison XL)	Negative
HBsAg antigen detection CLIA (DiaSorin Liaison XL)	Negative
Anti-HBs antibody detection CLIA (DiaSorin Liaison XL)	Negative
Anti-HBc antibody detection CLIA (DiaSorin Liaison XL)	Negative
Anti-HAV IgM antibody detection CLIA (DiaSorin Liaison XL)	Negative
Anti-HAV antibody detection CLIA (DiaSorin Liaison XL)	Negative
Anti-CMV IgM antibody detection CLIA (DiaSorin Liaison XL)	Negative
Anti-CMV IgG antibody detection CLIA (DiaSorin Liaison XL)	Positive
Anti-EBV VCA IgM antibody detection CLIA (DiaSorin Liaison XL)	Negative
Anti-EBV VCA IgG antibody detection CLIA (DiaSorin Liaison XL)	Positive
Anti-EBNA IgG antibody detection CLIA (DiaSorin Liaison XL)	Positive
Anti-HSV-1/2 IgM antibody detection ELISA	Negative
Anti-HSV-1/2 IgG antibody detection ELISA	Positive
Anti-VZV IgM antibody detection ELISA	Negative
Anti-VZV IgG antibody detection ELISA	Negative
Influenza A virus real-time PCR (Seegene)	Negative
Influenza B virus real-time PCR (Seegene)	Negative
Respiratory syncytial virus (RSV) detection real-time PCR (Seegene)	Negative
Adenovirus real-time PCR (Seegene)	Negative
Parainfluenza virus detection real-time PCR (Seegene)	Negative
Metapneumovirus real-time PCR (Seegene)	Negative
Rhinovirus real-time PCR (Seegene)	Negative
CMV real-time PCR (ELITe)	Negative
SARS-CoV-2 real-time PCR (GeneXpert)	Negative

## Data Availability

The original contributions presented in this study are included in the article. Further inquiries can be directed to the corresponding author.

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
