# Peer review of "Delayed Onset of Thrombotic Microangiopathy (TMA) upon Prolonged Carfilzomib Therapy in Multiple Myeloma: A Case Report and Comprehensive Review"

_pharmaceuticals, 2024, doi:10.3390/ph17121722_

Round 1
Reviewer 1 Report
Comments and Suggestions for Authors
Better discussion on the endothelial defects and the mechanisms induced by Carfilzomib.
Is there a glykocalyx defect/shedding?
Does complement attac metalloproteinases?
Defect glykocalyx impairs complement control ( Teoh CW, et al. The loss of glycocalyx integrity impairs complement factor H binding and contributes to cyclosporine-induced endothelial cell injury. Front Med (Lausanne). 2023 Feb 13;10:891513). Comment?
Did you analyse vWF and VWF/ADAMTS13 ratio? Comment?
Better discription on plasmepheresis procedures, it´s not enough just to mention 18. How long was each procedure? Volume of exchange? Every consecutive day? Was improvement satisfactory? Did you consider eculizumab at any stage? 18 days is a long period (or did you perform several plasmapheresis during the initial days? - with more extensive plasmaexchange in the initial phase?) in a critical ill patient. How do you evaluate a not satisfactory effect of plasmapheresis? Platelet level and or complement levels? Others?
Why did you use FFP and not TRALI safe Octaplas?
Author Response
Response to reviewer 1:
We want to thank the reviewer for his/her careful reading of our manuscript and thoughtful comments to improve it. In the following, we respond to the criticism and detail the changes we have made in our manuscript.
- Better discussion on the endothelial defects and the mechanisms induced by Carfilzomib.
We changed the introduction and the discussion sections to reflect better the role of endothelial injury in TMA caused by carfilzomib exposure.
- Is there a glycocalyx defect/shedding?Does complement attack metalloproteinases? Defect glykocalyx impairs complement control ( Teoh CW, et al. The loss of glycocalyx integrity impairs complement factor H binding and contributes to cyclosporine-induced endothelial cell injury. Front Med (Lausanne). 2023 Feb 13;10:891513). Comment?
Thank you very much for drawing our attention to the possible interaction of the complement system with the glycocalyx. Unfortunately, we did not have the chance to sample endothelial cells from our patient or to perform in vitro studies to elaborate on this hypothesis.
- Did you analyse vWF and VWF/ADAMTS13 ratio? Comment?
We measured VWF and ADAMTS13 activity from blood samples drawn just before the 1st PEX with the following results:
Von Willebrand factor antigen: 408%, VWF RCo: 219%, VWF CBA:462%, ADAMTS13 activity: 58%
VWF:Ag/ADAMTS13:Ac: 7 (very high)
VWF:CBA/ADAMTS13:Ac: 8 (very high)
The observed VWF:Ag level and VWF:Ag/ADAMTS13:Ac ratio is similar to those observed in 53 acute TMA patients with coexisting disease (malignancy, sepsis, heart surgery, solid-organ transplantation, systemic autoimmune disorders) in one of our previous studies (Farkas P et al, Immunobiology 2016, PMID: 27771173).
- Better discription on plasmepheresis procedures, it´s not enough just to mention 18. How long was each procedure? Volume of exchange? Every consecutive day? Was improvement satisfactory? Did you consider eculizumab at any stage? 18 days is a long period (or did you perform several plasmapheresis during the initial days? - with more extensive plasmaexchange in the initial phase?) in a critical ill patient. How do you evaluate a not satisfactory effect of plasmapheresis? Platelet level and or complement levels? Others?
High volume (median 1.4x plasma volume exchange, range 1.3-1.5; substitution fluid: 500 ml 4% human albumin + 14 units of FFP/session) daily PEX was immediately started when TMA was suspected. After excluding TTP and other causes of TMA, a request for complement inhibitory therapy insurance approval was submitted, and PEX therapy was continued. A complete TMA response (normalization of platelet count, LDH, haptoglobin + >25% decrease of serum creatinine) was achieved after 15 daily PEXs. As complement inhibitory therapy approval was still pending, three further daily treatments were performed, and PEX was finally stopped after 18 daily sessions without significant adverse events. We received the insurance approval for complement inhibitory therapy one week after terminating PEX treatment in remission. At that time, the patient was already well without any signs of TMA activity (normal platelet count, LDH, haptoglobin, sC5b-9 level, eGFR:55), and the approved eculizumab therapy became unnecessary.
- Why did you use FFP and not TRALI safe Octaplas?
In Hungary, fresh frozen plasma is the standard substitution fluid for TTP/aHUS/TMA indications. Octaplas is theoretically available, but we cannot use it due to its very high cost that is not reimbursed.
Reviewer 2 Report
Comments and Suggestions for Authors
This case study highlights the critical importance of vigilance for thrombotic microangiopathy as a rare but serious delayed complication of carfilzomib therapy in multiple myeloma patients. This report underscores the diagnostic challenges posed by the delayed onset of TMA, occurring 11 months after treatment initiation, and emphasizes the value of timely recognition and intervention. The successful management of this case, achieved through prompt drug discontinuation, plasmapheresis, and supportive care, reinforces the necessity of continuous monitoring for TMA throughout the course of carfilzomib therapy, regardless of duration, to improve patient outcomes and mitigate potential life-threatening complications. The work need to be revised be for publication as follows:
1. Avoid bulk citations like TMA, carfilzomib, a proteasome inhibitor approved for the treatment of multiple myeloma, has been identified as a potential causative agent 4-19. Keep only recent and most relevant one or two. Same way Line 206 should be checked for all citations.
2. Refer similar paper cite and mention how this work is differnet form that e.g. https://doi.org/10.1016/j.intimp.2024.112178. Improve the introduction with the addition of data on human clinical trials, Cross-Sectional Study (https://doi.org/10.3390/toxins15040293, https://doi.org/10.1016/j.prp.2024.155354)
3. Carfilzomib-induced TMA is a serious but rare adverse effect that typically presents within the first months of treatment initiation (20), add one or more reference to support this statement
4. Line 44-46 need support of some previous study. However, cases with a delayed onset, occurring several months after therapy commencement, are particularly rare and can complicate the clinical diagnosis and management due to their atypical presentation.
5. Method and case presentation data/sentences should be cited where ever application and generalized statements used for diagnosis support or proposed therapy.
6. Line 222-223: TMA associated with carfilzomib treatment represents a serious adverse event with 222 significant implications for patient management and outcomes 4-19. Few sentences are repeatable and rewritten in different way. Should be avoided. (Introduction similarity)
7. The author contribution section should be included.
Author Response
Response to reviewer 2:
We thank the reviewer for his/her valuable time spent to improve our manuscript through the peer review process. In the following, we summarize the changes in our manuscript as requested and respond item-by-item to the criticism.
- Avoid bulk citations like TMA, carfilzomib, a proteasome inhibitor approved for the treatment of multiple myeloma, has been identified as a potential causative agent 4-19. Keep only recent and most relevant one or two. Same way Line 206 should be checked for all citations.
Thanks for your comment, we deleted some of these references from the citation list to concentrate on the most appropriate ones. The introduction section was restructured accordingly,
- Refer similar paper cite and mention how this work is different form that e.g. https://doi.org/10.1016/j.intimp.2024.112178. Improve the introduction with the addition of data on human clinical trials, Cross-Sectional Study (https://doi.org/10.3390/toxins15040293, https://doi.org/10.1016/j.prp.2024.155354)
Thank you very much for drawing our attention to the important review paper on carfilzomib-induced thrombotic microangiopathy by Fang et al. In the revised text, we certainly cited this seminal paper. In this paper, most of the 66 cases were early-onset carfilzomib TMAs with a few exceptions. On the other hand, we think that with respect to the phenomena described in our paper, the relevance of microcystins (as described in Feng et al) and mesenchymal stem cells (as detailed in Liu et al) does not exist. These papers were not detailed in our introduction.
- Carfilzomib-induced TMA is a serious but rare adverse effect that typically presents within the first months of treatment initiation (20), add one or more reference to support this statement
We thank your suggestion and fully accept it, the seminal paper of Fang et al was cited here again.
- Line 44-46 need support of some previous study. However, cases with a delayed onset, occurring several months after therapy commencement, are particularly rare and can complicate the clinical diagnosis and management due to their atypical presentation.
The papers by Rassner et al ( DOI: 10.1186/s12882-020-02226-5), and Meseha et al (doi.org/10.1007/s00277-024-05965-9) were included as novel citations.
- Method and case presentation data/sentences should be cited wherever application and generalized statements used for diagnosis support or proposed therapy.
We changed our manuscript accordingly.
- Line 222-223: TMA associated with carfilzomib treatment represents a serious adverse event with 222 significant implications for patient management and outcomes 4-19. Few sentences are repeatable and rewritten in different way. Should be avoided. (Introduction similarity)
We corrected the repetitive sentences according to these suggestions.
- The author contribution section should be included.
We updated this section to better reflect the contributions of the individual authors.